# Constructing coherent spatial memory in LLM agents through graph rectification

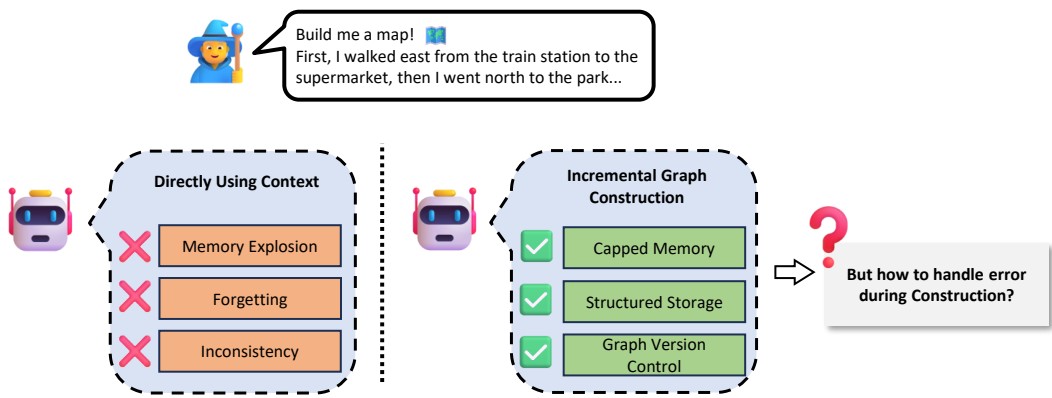

Figure 1: Spatial reasoning paradigms for LLM agents: Direct context-based approach with inherent limitations versus incremental graph construction with structured error handling capabilities.

## ABSTRACT

Given a map description through global traversal navigation instructions (e.g., visiting each room sequentially with action signals such as north, west, etc.), an LLM can often infer the implicit spatial layout of the environment and answer user queries by providing a shortest path from a start to a destination (for instance, navigating from the lobby to a meeting room via the hall and elevator). However, such context-dependent querying becomes incapable as the environment grows much longer, motivating the need for incremental map construction that builds a complete topological graph from stepwise observations. We propose a framework for LLM-driven construction and map repair, designed to detect, localize, and correct structural inconsistencies in incrementally constructed navigation graphs. Central to our method is the Version Control, which records the full history of graph edits and their source observations, enabling fine-grained rollback, conflict tracing, and repair evaluation. We further introduce an Edge Impact Score to prioritize minimal-cost repairs based on structural reachability, path usage, and conflict propagation. To properly evaluate our approach, we create a refined version of the MANGO benchmark dataset by systematically removing non-topological actions and inherent structural conflicts, providing a cleaner testbed for LLM-driven construction and map repair. Our approach significantly improves map correctness and robustness, especially in scenarios with entangled or chained inconsistencies. Our results highlight the importance of introspective, history-aware repair mechanisms for maintaining coherent spatial memory in LLM agents.

## 1 INTRODUCTION

LLMs have shown strong abilities in open-domain reasoning, sequential planning, and text-based navigation. However, in text-processing environments, spatial cognition with LLMs still primarily depends on direct reasoning within the context window (Ding et al., 2024). However, this approach presents several potential challenges: it may exceed context capacity limitations when processing

extensive texts, encounter context forgetting issues when addressing complex problems, and introduce inconsistencies in iterative reasoning processes (see Figure 1). Consequently, when tackling complex large-scale spatial problems, adopting a human-like cognitive approach—progressively assembling local spatial cognition to achieve understanding of complex spaces (Xia et al., 2025)—may constitute a superior solution. For LLMs, this methodology alleviates contextual pressure by incrementally storing local spatial cognition in graph structures, requiring the context to process only current local information. Furthermore, the structured storage through graph representation ensures consistency in search structures and provides error correction capabilities when cognitive biases occur. Despite this potential, the question of how to effectively represent and maintain complex spatial layouts in such graph structures remains largely unexplored. In particular, small perceptual or reasoning errors made early in the mapping process can silently propagate through the structure, eventually leading to severe inconsistencies.

While structural conflicts such as duplicated node names in different location or multiple outgoing edges in the same direction can be detected once they appear in the graph, the *underlying errors* that cause these conflicts, such as incorrect edge insertions or misnamed nodes — may have been introduced much earlier. These causal errors often remain unnoticed until enough context has accumulated in the graph; only then do they emerge as inconsistencies. This temporal gap between the cause and the observable conflict makes diagnosis and repair especially challenging.

Worse still, many of these errors exhibit *coupled dependencies*: A single erroneous edge or misinterpretation of spatial relationships can trigger a cascading sequence of incorrect additions, creating interconnected chains of errors that cannot be resolved through examination of the current graph state alone. Since most LLMs lack persistent memory or Version Control, they are poorly equipped to trace the provenance of errors or reason about when and why a faulty edge was introduced.

To address this challenge, we propose **LLM-MapRepair**, a modular framework for detecting and repairing topological inconsistencies in navigation graphs constructed by LLM agents. At the core of our method is the **Version Control**, a versioned graph history that records every modification to the graph, along with its originating observation and time indexed head. Version Control enables time-aware tracing, rollback, and structural comparison, allowing the system to pinpoint the specific actions that introduced inconsistencies, even if they occurred many steps earlier.

To improve the efficiency of graph repair, we introduce an **Edge Impact Score** to prioritize repair actions by estimating the potential downstream effects of each edge based on reachability, usage frequency, and conflict propagation. This enables the system to identify low-impact edges that can be safely edited or removed, thereby reducing the risk of introducing further inconsistencies during the repair process.

We evaluate our approach on environments from the MANGO benchmarks (Ding et al., 2024), where LLM agents construct navigation graphs from raw textual observations. Experiments show that our method significantly improves structural integrity and overall task performance, especially in cases involving long-range error propagation. Our contributions are as follows:

- We identify a critical limitation of LLM-based agents in long-horizon exploration: their inability to detect and correct accumulated structural errors that emerge from temporally distant actions.
- We propose a history-aware graph repair framework, integrating Version Control-based error tracing, Edge Impact scoring.
- We refine the MANGO (Ding et al., 2024) benchmark dataset by systematically removing all non-topological actions and inherent structural conflicts, creating a topologically consistent dataset better suited for evaluating LLM-based spatial mapping and navigation.

## 1.1 RELATED WORK

**Enhancing LLMs Spatial Reasoning.** Recent advances have improved LLMs' spatial reasoning through specialized training and prompting. AlphaMaze (Dao & Vu, 2025) combines supervised learning with reinforcement learning (GRPO) for maze navigation, while Mind's Eye (Wu et al., 2024b) employs "visualization-of-thought" prompting to simulate internal spatial imagery. Despite these improvements, both approaches remain fundamentally limited by the model's contextual capacity and lack mechanisms to maintain consistency across extended spatial reasoning tasks.

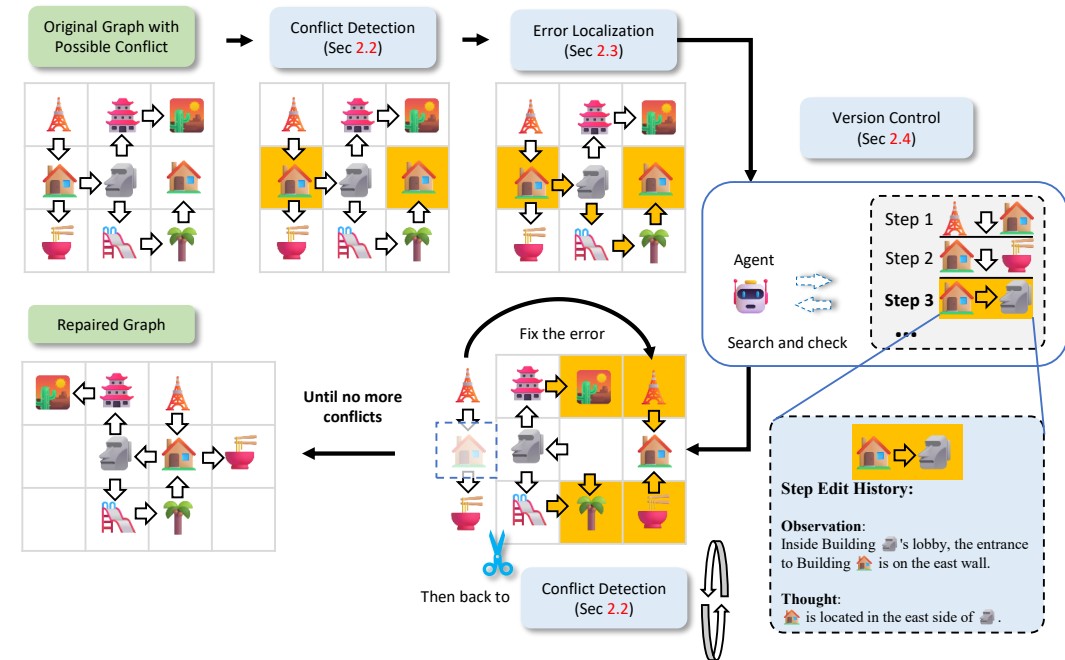

Figure 2: Overview of the LLM-MapRepair framework.

**Evaluation of Map Construction in Language Agents.** While LLMs demonstrate local spatial understanding, they fail to maintain consistent global world models during incremental exploration. The MANGO benchmark (Ding et al., 2024) reveals that even GPT-4 struggles with long-distance planning and localization under partial observations, with performance degrading beyond 70 reasoning steps due to context limitations. Modular navigation frameworks (Zhang & Ji, 2025) attempt to address this through planning-execution modules but critically lack graph-level consistency tracking, leaving structural inconsistencies undetected and uncorrected during extended exploration.

**Spatial Representations and Vision assisted Scene Graphs.** Explicit graph representations have shown promise for structured navigation. SG-Nav (Yin et al., 2024) constructs 3D scene graphs for zero-shot object navigation, UniGoal (Yin et al., 2025) provides unified multi-modal graph representations, and VoroNav (Wu et al., 2024a) uses Voronoi-based spatial partitioning. However, these methods suffer from two critical limitations: they assume static environments where graphs remain unchanged, and they rely heavily on pre-built visual scene graphs, making them unsuitable for dynamic text-based exploration where errors accumulate over time. Existing approaches fail to address the fundamental challenge of maintaining graph consistency during dynamic construction from textual input. Current methods lack systematic error detection and repair mechanisms, leading to compounding inconsistencies that degrade performance over extended exploration sequences. Our work fills this gap by introducing a comprehensive framework for graph construction and repair that operates solely on textual descriptions while actively detecting and correcting structural inconsistencies as they emerge.

## 2 APPROACH

### 2.1 OVERVIEW OF THE FRAMEWORK

As mentioned in Sec 1, directly using LLM context to process complex textual scenarios is impractical. For instance, the spirit game in the MANGO dataset contains up to 1,264 observation steps, a text length that greatly exceeds LLM capabilities. Consequently, when Ding et al. (2024) tests the MANGO dataset, the authors only evaluated the first 70 steps of each game. To enable LLMs to handle complex long texts, we introduce an incremental graph construction approach. The LLM incrementally updates a graph by recording spatial relationships from each new observation and

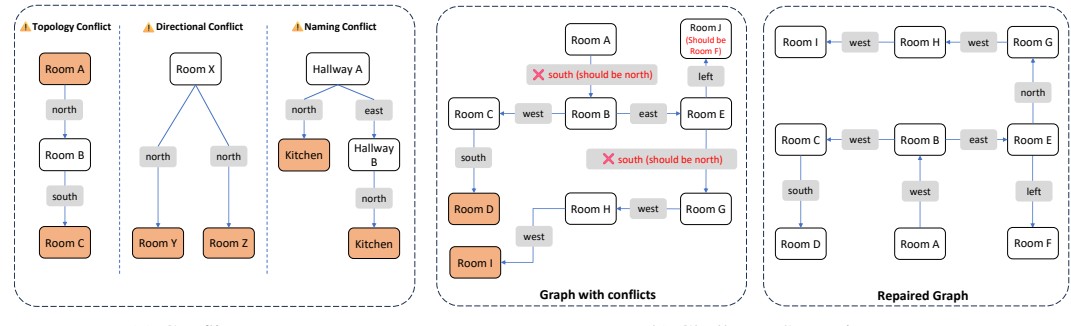

(a) Conflict Types          (b) Challenge Scenario

Figure 3: (a) Three types of structural conflicts — naming conflict, directional conflict, and topological conflict. Highlighted nodes indicate the conflicting pairs. (b) A challenging conflict scenario: a misdirected edge from Room E to Room G introduces a latent spatial misalignment. A visible topology conflict emerges later between Room D and Room I (highlighted), while correcting the edge E→G may trigger a new conflict between Room H and Room A. The edge E→J is incorrect, though it is unrelated to the current conflict.

automatically merging them into a complete graph. To ensure the structural consistency of topological maps incrementally constructed by LLMs in partially observable environments, we propose a modular repair framework that detects and corrects errors as they emerge during exploration.

Figure 2 illustrates the complete workflow of the LLM-MapRepair framework for spatial graph construction and repair. The framework operates cyclically to transform conflict-laden graphs into structurally consistent representations through systematic error detection and targeted corrections.

The repair process consists of three integrated stages that work in concert. Conflict Detection (Sec 2.2) systematically identifies structural inconsistencies within the graph, highlighting problematic connections that violate spatial constraints. Error Localization (Sec 2.3) then analyzes these detected conflicts, employing an Edge Impact Scorer to prioritize high-weight erroneous edges and trace their origins to specific reasoning steps. The Version Control component (Sec 2.4) maintains comprehensive historical context through a Version Control module, preserving both original observations and the reasoning processes that generated each edge.

The framework's strength lies in its introspective capabilities. When examining the commit history, the agent can access complete contextual information, including original observations ("Inside Building is lobby, the entrance to Building is on the east wall") and associated reasoning thoughts ("is located in the east side of"). This historical transparency enables the agent to identify flaws in original spatial reasoning and apply targeted repairs. After each "Fix the error" operation, the system returns to Conflict Detection to verify that corrections haven't introduced new inconsistencies, continuing until a conflict-free "Repaired Graph" is achieved. By maintaining versioned graph history and performing targeted, low-impact corrections, this framework enables robust introspective reasoning over dynamically constructed maps, proving especially effective in long-horizon exploration scenarios where early errors may only manifest during later stages of the process.

## 2.2 CONFLICT DETECTION

As LLM agents build navigation graphs from text, inconsistencies may gradually accumulate, resulting in structural conflicts. We identify three major types—*naming*, *directional*, and *topological*—as illustrated in Figure 3(a) conflict types:

- **Topological Conflict:** Arises from invalid graph structures such as cycles in tree-like spaces, unreachable nodes, or over-connected components. *Detection:* Use graph traversal, cycle detection, and connected component analysis.

- **Directional Conflict:** Happens when a node has multiple outgoing edges labeled with the same direction (e.g., two "north" edges), which violates spatial constraints. *Detection:* Enforce a single outgoing edge per direction for each node.

- **Naming Conflict:** Occurs when different locations are assigned the same name (e.g., two "Kitchens" in distinct positions), leading to ambiguity in reasoning and localization. *Detection:* Check for identical names associated with distinct coordinates using name hashing.

## 2.3 ERROR LOCALIZATION

Resolving conflicts in LLM-generated navigation graphs is often more difficult than detecting them, due to several intertwined challenges illustrated in Figure 3(b). The primary complexity stems from **delayed conflicts**, where errors introduced early may not be noticed until much later in the exploration process. As demonstrated in the challenge scenario, a wrong direction from *Room E to G* leads to a cascade of misplacements, but the actual conflict only becomes apparent when the loop reaches overlapping *Rooms D and I*. This temporal gap between error introduction and detection is further complicated by **entangled conflicts**, where attempting to fix one edge can inadvertently create new conflicts elsewhere in the graph. For instance, adjusting the *Room E→G* connection resolves the initial overlap but simultaneously causes a new conflict between *Rooms H and A*. Perhaps most problematically, **silent errors** can persist undetected due to the absence of contradictory evidence. The incorrect direction from *Room E to J* exemplifies this issue, causing no immediate conflict while silently corrupting the underlying map structure.

The asynchrony between graph construction errors and structural conflicts motivate our framework's separation of **conflict detection** and **error localization**, enabling robust identification of true error sources through temporal and structural reasoning.

Once a conflict is detected, the system must identify not just the conflicting edges, but the actual root cause that introduced the inconsistency. This is non-trivial, as the erroneous edge may lie far from the observed conflict and may even appear structurally correct in isolation. Our localization pipeline proceeds in four stages: (1) identifying the minimal conflicting path pair, (2) computing their lowest common ancestor (LCA), (3) extracting divergent edges as error candidates, and (4) scoring and ranking these candidates by impact.

**Minimal Conflicting Path Pair** Given a structural conflict (e.g., naming or topology), we first locate two distinct paths that lead to the conflicting nodes. For instance, in Figure 3 challenge scenario, the topology conflict between Room D and Room I can be traced to two paths:

$$\text{Path}_1 : \text{Room B} \rightarrow \text{C} \rightarrow \text{D}$$
$$\text{Path}_2 : \text{Room B} \rightarrow \text{E} \rightarrow \text{G} \rightarrow \text{H} \rightarrow \text{I}$$

Both paths result in overlapping node positions, violating spatial exclusivity constraints.

**Lowest Common Ancestor (LCA)** To identify where the error first diverged, we compute the lowest common ancestor (LCA) of the two conflicting paths. This is the last node shared between them before the divergence that leads to inconsistency. Formally, for paths $p_1$ and $p_2$, let:

$$\text{LCA}(p_1, p_2) = \max\{v \mid v \in p_1 \cap p_2,\ \text{order}(v) \text{ is minimal}\} \tag{1}$$

In Figure 3, Room B serves as the LCA. Edges beyond this node are considered candidate error sources.

**Candidate Edge Extraction** We extract the divergent subpaths from the LCA to each conflict node and collect all edges along these subpaths as potential causes of the inconsistency. In the example:

$$\text{Candidate edges} = \{\text{B} \rightarrow \text{C},\ \text{C} \rightarrow \text{D},\ \text{E} \rightarrow \text{G},$$
$$\text{G} \rightarrow \text{H},\ \text{H} \rightarrow \text{I}\}$$

Additionally, silent errors (e.g., E→J) not yet resulting in conflicts can also be included as fallback candidates for global ranking.

**Edge Scoring and Ranking** To determine which candidate edge to prioritize for inspection and repair, we assign each edge a composite score based on three factors: *reachability*, *conflict count*, and *usage*. These reflect the potential structural impact, the degree of inconsistency evidence, and the reliance of observed paths on the edge, respectively.

PageRank-Based Heuristic Motivation. We draw theoretical motivation from PageRank (Brin & Page, 1998), which models the importance of a node as the stationary distribution of a random walk. Extending this idea to edges, we treat error propagation as a stochastic process over the graph and model the *importance* of an edge in contributing to potential error spread and repair cost. Specifically, we define three factors, each reflecting a different mode of edge influence:

- **Reachability** captures the *structural influence* of an edge: the number of downstream nodes reachable from edge $e$, reflecting how far an error at $e$ could theoretically propagate in the graph.
- **Conflict Count** captures the *error-generating potential* of an edge: the number of distinct conflicts in which $e$ participates, serving as direct evidence of its contribution to observed inconsistencies.
- **Usage** captures the *contextual dependency* of an edge: the number of conflict-related paths that include $e$, indicating how often it is actually relied upon in failure scenarios. While usage is constrained by reachability, they capture distinct dimensions—reachability reflects global potential, whereas usage reflects local empirical relevance.

According to the convergence properties of the PageRank process, a uniform combination of influence features—without weighting—is sufficient to approximate relative importance under general assumptions. To avoid introducing hard-to-tune hyperparameters, we adopt a simple unweighted scoring function, after min-max normalization within the candidate set:

$$\text{score}(e) = \widehat{\text{Reach}}(e) + \widehat{\text{Conflict}}(e) + \widehat{\text{Usage}}(e) \tag{2}$$

**Repair Prioritization Objective**  Edge scoring not only identifies likely error sources, but also informs which edge to inspect or repair first. We prioritize edges that can either resolve existing conflicts or trigger new ones—thereby reducing ambiguity and accelerating convergence.

To formalize this, we define the expected *Conflict Revelation Gain* (CRG) of an edge $e$ as:

$$\text{CRG}(e) \triangleq \mathbb{E}\left[|\mathcal{C}_{t+1}| - |\mathcal{C}_t| \,\middle|\, \text{repair}(e)\right], \tag{3}$$

where $\mathcal{C}_t$ is the set of known conflicts at time $t$. A high-CRG edge is likely to expose hidden errors or clarify causal paths.

While CRG is hard to compute directly, our score approximates it:

$$\text{score}(e) = \widehat{\text{Reach}}(e) + \widehat{\text{Conflict}}(e) + \widehat{\text{Usage}}(e), \tag{4}$$

capturing structural influence, inconsistency evidence, and observed dependency.

Edges are thus prioritized by descending score. However, to trace an edge's origin, estimate its downstream effects, or reverse a mistaken fix, we must maintain temporal structure over graph edits—this motivates the next component: the Version Control.

## 2.4 VERSION CONTROL

To support long-term consistency in LLM-driven graph construction, we introduce the Version Control—a lightweight, structured history mechanism that logs all changes to the navigation graph across time. Unlike flat logs or linear event lists, Version Control maintains versioned snapshots of the graph in a directed chain, enabling targeted rollback, difference analysis, and recall thinking history.

**Version Control Structure.** Version Control is a directed chain of version records $[G_0, G_1, \ldots, G_t]$, where each commit history $G_i$ represents a step-wise change to the graph. Rather than storing full graph snapshots, each version logs only the incremental updates. (See Figure 4)

$$G_i = \{\texttt{Step\_id}, \texttt{Commit}, \texttt{Trigger\_event}, \texttt{Observation\_id}, \texttt{Analysis}\}$$

This structure minimizes memory cost while enabling exact reconstructions.

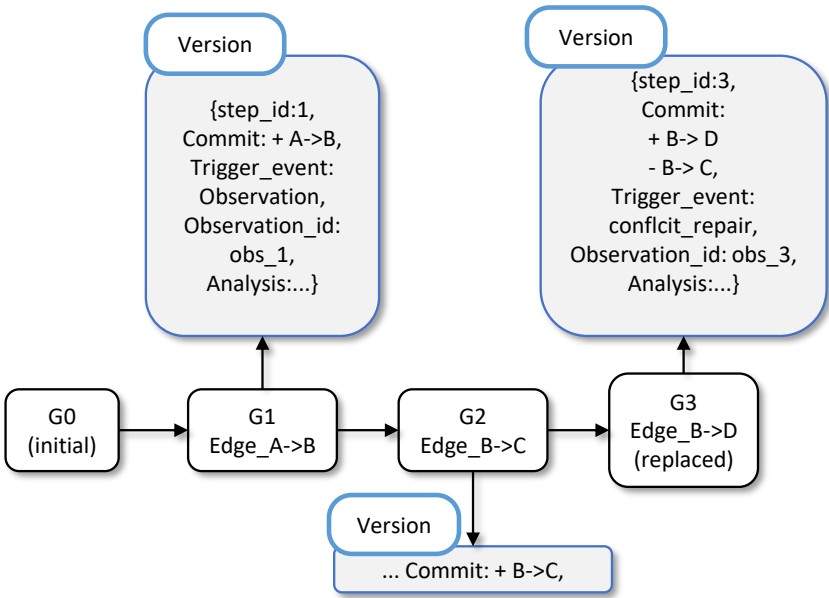

Figure 4: Each commit $G_i$ represents a commit to the graph (e.g., edge additions or conflict-triggered replacements). The Version Control system maintains a linear commit history where each commit is indexed by its corresponding edge modification. Commit metadata includes the step identifier, specific edge changes (+ for additions, - for removals during replacements), trigger event type, and associated observation. The system supports three core operations: rollback to previous commits, recall of commit details, and diff comparisons between commits.

**Supported Operations.** Version Control supports three key operations:

- `rollback_to(version)`: Restores the graph to a prior state by undoing subsequent steps.

- `recall_step(version)`: Obtain the thinking history corresponding to the step.

- `diff(G_i, G_j)`: Computes edge-level differences between two versions.

These operations support both runtime repair decisions and post hoc analysis.

**Incremental Evolution.** Every LLM-initiated interaction—whether through new observations or repair actions—triggers a graph update and logs a new version in Version Control. This guarantees that even failed or partially correct decisions are preserved for future analysis. Version Control records whether an update was conflict-triggered (e.g., `Trigger_event = conflict_repair`) to provide interpretability in version history. Version Control enables following key capabilities:

- **Graph alignment**: Compare versions before and after a repair to assess changes.

- **Structural diffing**: Detect which steps introduced regressions or inconsistencies.

- **Error propagation tracing**: Model how errors spread over time.

Unlike flat logs or event lists, the versioned graph design supports non-destructive rollbacks and dependency-aware repair strategies.

This aligns with well-established principles in database systems, where *write-ahead logging (WAL)* (Gray & Reuter, 1993) ensures recoverability and traceability by recording all state changes. Similarly, Version Control gives LLM-based systems a foundation for Version Control, self-debugging, and structured repair—all essential for interactive, persistent reasoning tasks.

## 3 EXPERIMENT

### 3.1 DATASET

We conduct our experiments on the MANGO benchmark (Ding et al., 2024), a curated collection of 53 interactive fiction (IF) environments originally derived from the Jericho benchmark (Hausknecht et al., 2019). Unlike Jericho, MANGO excludes non-spatial actions (e.g., "take" or "examine"), ensuring that every action directly corresponds to a location change. This design choice makes the environment ideal for evaluating the construction and repair of navigational graphs.

Each episode in MANGO consists of an agent performing a series of movement commands (e.g., "go north", "go down") based on textual observations. The resulting action-observation trajectory is used to incrementally build a topological map. However, we found that the original dataset itself contains numerous structural conflicts and non-topological actions, which requires us to first correct the graphs. The detailed steps for removing structural conflicts can be found in the appendix.

### 3.2 GRAPH CONSTRUCTION PROCESS

We employ an LLM to incrementally construct navigation graphs by processing each step of the walkthrough sequentially. For each game, the LLM reads the step-by-step walkthrough and builds the navigation graph incrementally, where each action becomes an edge in the graph and each location becomes a node. The LLM only creates new edges or nodes when it determines that the current location has changed based on the textual observations.

Table 1: Comparison of Repair Method Performance

| Method | Avg. Loops | Repair Rate (%) | Accuracy (%) |
|---|---|---|---|
| Edge-Impact Ranking Only | **6.39** | **75.21** | 44.69 |
| Version Control Only | 7.44 | 63.03 | 54.00 |
| **Version Control+Edge-Impact Ranking** | 8.20 | 68.91 | **54.88** |
| Baseline(GPT-4o) | 9.52 | 21.85 | 5.77 |

### 3.3 ABLATION STUDY

To evaluate the contribution of different components in our repair framework, we conduct an ablation study based on the conflict-prone graphs identified in Table 1. We introduce the LLM-based graph repair loop with different tool configurations to measure their individual impact on repair performance. During the repair process, each conflict is given a maximum of 10 repair attempts. If secondary conflicts arise during the repair process, they do not consume additional repair opportunities. Throughout the entire conflict resolution process, the LLM maintains context containing historical repair information to inform subsequent decisions. In all experiments, the LLM is GPT-4o.

We compare four settings in Table 1: Edge-Impact Ranking Only, which prioritizes repair candidates by their scores; Version Control Only, which relies solely on the Version Control for history and rollback; Version Control + Edge-Impact Ranking, our full method combining both; and a Baseline without filtering or prioritization.

The table shows three key metrics for each method: *Avg. Loops* (average number of repair iterations required), *repaired* (number of conflicts successfully addressed), and *correct* (number of conflicts resolved with correct solutions).

The results in Table 1 reveal distinct performance characteristics for each repair strategy. The **Edge-Impact Ranking Only** method achieves the lowest average repair iterations (6.39 loops), requiring approximately 14% fewer iterations compared to Version Control-only and 33% fewer than the baseline approach. This efficiency stems from its focus on edge dependencies, preferring to quickly modify edges that are likely to trigger secondary conflicts, thereby reducing the total number of modifications needed. With a repair rate of 75.21% (179/238), the method successfully addresses 19% more conflicts than Version Control-only and over 240% more than the baseline. However,

while Edge Impact Ranking excels at conflict resolution, its accuracy is only 44.69%—fixing conflicts does not necessarily mean the underlying graph errors are properly corrected, as conflicts and actual errors are not strongly correlated.

The **Version Control-only** approach exhibits higher average loop counts (7.44, approximately 16% more than Edge Impact Ranking) due to the additional operations it performs, such as rollback actions and edge information queries, each consuming iteration cycles. However, Version Control's access to historical context and reasoning information recorded during edge insertion enables more accurate identification of root causes. With an accuracy of 54.00% (81/150), it achieves 21% higher accuracy than Edge Impact Ranking despite a lower repair rate of 63.03%. This is particularly effective when the candidate edge set is small, leading to more reliable corrections.

When combining both approaches (**Version Control + Edge-Impact Ranking**), we observe a synergistic effect. While the average loop count increases to 8.20 (28% higher than Edge Impact Ranking alone), the accuracy improves to 54.88% (90/164). This represents a 22.8% relative improvement in accuracy over Edge Impact Ranking alone, while maintaining a reasonable repair rate of 68.91%. The combined method leverages both structural impact analysis for efficient candidate identification and temporal context for accurate root cause analysis.

The **Baseline(GPT-4o)** without candidate filtering or prioritization mechanisms performs poorly across all metrics. With an average of 9.52 loops per repair attempt, it achieves only a 21.85% repair rate (52/238) and a mere 5.77% accuracy (3/52). Most conflicts reach the maximum iteration limit without successful resolution, highlighting the critical importance of structured error localization and prioritization in graph repair tasks.

We also embed **Version Control + Edge-Impact Ranking** into other LLM modules (e.g., GPT-4o-mini, GPT-4.1) to verify the generalization of our method, please refer to supplementary material for more details. Using the game "event" from Mango benchmark as input, an example of graph construction and repairing is appended to showcase the result from our method.

## 4 SUMMARY & LIMITATIONS

We present a framework for repairing navigation graphs constructed by LLms during exploration. While LLMs can incrementally build topological maps from language observations, their outputs are often noisy—introducing misaligned edges, duplicates, or subtle conflicts that accumulate and degrade reasoning.

To address this, we propose a three-stage repair pipeline: conflict detection, error localization, and impact-aware correction. Central to this is the Version Control, a versioned history of the map that supports rollback, difference analysis, and causal tracing of errors. We also introduce an Edge Impact Score to prioritize low-risk edits by quantifying structural and usage-based influence.

Experiments on grid-like maps show significant gains in reliability, though challenges remain in generalizing to dynamic environments, refining heuristic-based edge ranking, and detecting silent errors beyond visible conflicts. These results highlight the importance of making LLM agents not only build but also check and repair their evolving world models.

## 5 THE USE OF LARGE LANGUAGE MODELS (LLMS)

In this paper, LLMs were employed solely as a tool for text refinement and did not contribute to the conceptualization of the work.

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
