# OpenReview forum: "Constructing coherent spatial memory in LLM agents through graph rectification"
_ICLR.cc/2026/Conference — ICLR 2026 Conference Withdrawn Submission_

### Official Review · Reviewer_eVg9 · 2025-10-26

**Soundness:** 2
**Presentation:** 2
**Contribution:** 2
**Rating:** 6
**Confidence:** 3

**Summary:**

This paper presents a novel framework, LLM-MapRepair, for enabling large language model (LLM) agents to build and maintain coherent spatial memory during textual navigation tasks.
Unlike prior works that rely solely on LLM contextual reasoning, the proposed approach incrementally constructs a topological graph of the environment and introduces mechanisms for error detection, localization, and repair.

**Strengths:**

1. Unlike prior LLM-based navigation works that depend purely on context-window reasoning, this study formalizes a persistent, structured memory mechanism that records graph evolution and enables rollback and repair.
2. Current LLMs cannot sustain consistent spatial understanding over long reasoning horizons due to context limits and forgetting.
Emulating human-like incremental mapping and introducing self-repair mechanisms are convincingly argued as necessary steps for robust embodied reasoning.

**Weaknesses:**

1. Evaluation is restricted to text-based static environments; lacks validation in dynamic or multimodal contexts.
2. The link to embodied or real-world robotic tasks could be emphasized more clearly.
3. The evaluation domain is narrow, so the generality of the proposed approach for other tasks (e.g., visual navigation, embodied dialogue) remains uncertain.
4. The lack of definition or quantitative analysis of map scale. The paper never specifies how many nodes or edges each constructed navigation graph typically contains, nor does it provide any complexity analysis or scaling study.
As a result, it remains unclear how large a map the proposed framework can effectively handle before memory usage, repair cycles, or LLM context length become bottlenecks.

**Questions:**

See Weaknesses*

---

### Official Review · Reviewer_NmKs · 2025-10-29

**Soundness:** 2
**Presentation:** 2
**Contribution:** 4
**Rating:** 2
**Confidence:** 5

**Summary:**

The authors propose a framework for repairing navigation graphs constructed by LLMs.

**Strengths:**

The authors have a really great idea via-a-vis their framework which in addition to their untended use for repair, can also be used to diagnose problems in LLM navigation.

**Weaknesses:**

*Figure 2 needs more explanation
* the authors propose a framework and then run an experiment that is confusing and not clear what they hope to accomplish. I recommend that in future work, they focus on showing why LLMs perform best/works across their 3 sections and also do some sensitivity analyses on their Edge Impact Score, particularly in light of complexity.
*They also mention they want to enable LLMs to handle complex long texts, but I would focus on this after they have a more stable approach.

**Questions:**

*Why would someone need to use your framework?
*how could the results be used for future improvements?
*What are the limitations of your framework?
*How sensitive is your approach/scores to prompt variability?

---

### Official Review · Reviewer_Wcjx · 2025-11-01

**Soundness:** 2
**Presentation:** 1
**Contribution:** 2
**Rating:** 2
**Confidence:** 3

**Summary:**

This work proposes LLM-MapRepair, a framework that enables large language models to build and maintain consistent spatial maps from textual navigation instructions. Instead of relying solely on context reasoning (which leads to memory limits and inconsistencies) the method incrementally constructs a topological graph and repairs it through three integrated stages: conflict detection, error localization, and version-controlled repair. A central contribution is the introduction of a Version Control mechanism that rcords every graph edit for rollback and causal tracing, and an Edge Impact Score that prioritizes low-risk repairs based on reachability and path usage. Evaluated on a refined version of the MANGO benchmark, the approach significantly improves map correctness and robustness, especially for long-horizon navigation with accumulated inconsistencies.The experiments highlight the importance of history-aware reasoning for maintaining coherent spatial memory in LLM agents

**Strengths:**

The paper:
- Clearly identifies and motivates the problem of inconsistent spatial memory in LLMs
- Proposes a novel and inventive solution: a modular andd interpretable framework (LLM-MapRepair) for detecting and fixing map inconsistencies
- Introduces Version Control for persistent, history-aware reasoning, enabling rollback and causal tracing
- Defines an Edge Impact Score to prioritize low-risk, high-impact repairs using a principled heuristic
- Provides strong experimental validation with clear ablations showing complementary effects of each module
- Cleans and refines the MANGO benchmark, improving evaluation quality for future work
- Some of the figures make the approach easier to follow and conceptually well-motivated

**Weaknesses:**

- The paper is hard to read in its current form. Some examples below:
   - The paper uses not-so-common terms without defining them, e.g., “local spatial cognition,” “cognitive biases,” “structural integrity,” and “contextual pressure.”
   - In the introduction, the solution appears before the setup: what exactly are nodes and edges? I was only able to figure it out after reading the methods section.
- Problem formulation and evaluation criteria are not specified clearly.
- The motivation for using LLMs (as opposed to simpler map builders) is not provided.
- Evaluated on a single dataset, and method appears tailored to it.
- No baseline comparisons beyond GPT-4o, making it difficult to assess the method’s relative performance.
- The paper lacks a deeper analysis explaining the method’s behavior, limitations, or challenges in solving the problem.

**Questions:**

- Why do we need an LLM for this mapping problem at all? What capabilities are essential here?
- The paper motivates with "human-like cognition", but what evidence suggests human spatial memory is graph-based?
- Fig. 1 mentions forgetting in the LLM context; why would information in-context be "forgotten"?
- If graph errors stem from LLM reasoning, how does version control fix them? Maybe shortening the reasoning context? Please motivate more clearly.
- L158: How does 1k observation steps exceed context? What's the average tokens per observation?
- The conflict taxonomy is confusing; aren't directional conflicts fundamentally topological?
- "Delayed/entangled/silent" conflicts are introduced, but how is it tied to the solution later?
- Why is "usage" a factor in resolving edge conflicts if all conflicts will be resolved anyway? How is it computed?
- Why assume the true error lies between detected conflicts and their LCA?
- The paper mentions Conflict Revelation Gain, but replaces it with a score function defined earlier. Why is that a good approximation?
- How are each individual component of the score computed?

---

### Note · Authors · 2025-11-27

I have read and agree with the venue's withdrawal policy on behalf of myself and my co-authors.